# CHANNEL-SIMILARITY AWARE SPIKE ENCODING FOR MULTIVARIATE TIME-SERIES FORECASTING

## ABSTRACT

Spiking Neural Networks (SNNs) have attracted increasing attention for multivariate time-series forecasting due to their intrinsic energy efficiency and suitability for modeling temporal signals. However, existing SNN-based studies have largely focused on temporal dynamics, restricting their scope to spike encoding and temporal modeling. In contrast, recent advances in artificial neural networks demonstrate that modeling inter-channel similarity can substantially enhance forecasting performance. Despite this, such perspectives remain underexplored in the context of SNNs. In this work, we introduce a method that explicitly models inter-channel similarity through channel clustering and integrates it into temporal spike encoding. Specifically, we employ attention-based clustering to quantify channel similarity as cluster memberships, and leverage the Straight-Through Estimator (STE) to enable both end-to-end optimization and seamless integration into spike-based encoding. We evaluate the proposed approach on six benchmark datasets spanning diverse domains and temporal characteristics, using recurrent-, transformer-, and convolution-based SNN backbones. Experimental results show consistent improvements over baseline SNNs, achieving relative reductions in RRSE ranging from 3.0% to 6.5%. These findings highlight the potential of inter-channel similarity modeling as a complementary dimension to temporal dynamics in advancing the forecasting capabilities of SNNs.

## 1 INTRODUCTION

Spiking Neural Networks (SNNs) are third-generation neural networks that emulate the behavior of biological neurons by representing information as discrete spike signals and performing computations only when events occur. Through membrane potential accumulation, firing, and leakage, SNNs process temporally continuous signals while offering significantly higher computational efficiency and lower energy consumption than conventional Artificial Neural Networks (ANNs). In particular, SNNs operate with extremely low power consumption on neuromorphic platforms, e.g., Intel Loihi 2 (Orchard et al., 2021) and IBM TrueNorth (Akopyan et al., 2015), making them attractive for wearable sensors, IoT devices, and other resource-constrained settings. Furthermore, spike-based computation directly reflects the temporal intervals between input events, providing structural advantages when handling time-series data. Accordingly, SNNs are increasingly recognized as promising predictive models for various time-dependent tasks (Sharma & Srinivasan, 2010; Laña et al., 2018; 2019; Zhou et al., 2021b; Ibad et al., 2022). Time-series forecasting itself is now fundamental in numerous domains such as financial market analysis (Chaudhary, 2025; Sonani et al., 2025), energy demand prediction (Mishra et al., 2024; Orji et al., 2025), traffic flow forecasting (Qin et al., 2025), and weather prediction (Akshaya et al., 2024), with numerous ANN-based models (Liu et al., 2024; Chen et al., 2024b; Dai et al., 2024) achieving state-of-the-art performance. A key finding in recent work is that explicitly modeling inter-channel interactions can substantially improve forecasting accuracy for multivariate time series. Consequently, attention-based fusion (Zhang & Yan, 2023; Ilbert et al., 2024) and channel clustering (Chen et al., 2024a; Qiu et al., 2025) have been proposed. However, these ANN-based approaches rely on complex weight calculations and dense interaction structures, incurring high computational and energy costs, which limit their applicability in edge or battery-powered environments. SNN-based forecasting seeks to alleviate this burden through spike encoding techniques (Banerjee et al., 2022; Lv et al., 2024b) and spiking neuron architectures (Feng et al., 2025). Although recent SNN models achieve competitive accuracy

with light-weight computation, they focus almost exclusively on temporal dynamics—such as encoding time information and regulating neuronal firing patterns and typically process each channel independently. Because classical spike encoders lack a mechanism for representing inter-channel relationships, salient dependencies in multivariate data remain under-exploited, limiting forecasting accuracy. To address the above limitations, we propose a channel-similarity-aware spike encoding method that embeds inter-channel similarity information into SNN-based forecasting models in a spike-compatible form. The method first applies an attention-based channel clustering—previously shown effective in ANN literature—to quantify pairwise interactions among channels. This structural information is then converted, via a Straight-Through Estimator (STE), into learnable binary spike representations, which are subsequently injected into the temporal encoding stage. Through this integration, the proposed framework enables an SNN to exploit temporal dynamics and channel-wise structural information concurrently within a unified spiking architecture. The main contributions of this work are summarized as follows:

- Spike-form channel similarity integration: We propose the first approach that explicitly transforms inter-channel similarity into a spike-compatible format and integrates it into SNN forecasting models, enabling the use of structural information without violating the sparse binary computation paradigm of SNNs.

- Modular design and high applicability: The proposed method can be easily integrated into a wide range of SNN backbones and encoding methods, such as Spike-RNN and Spikformer, and we demonstrate its performance improvements and extensibility through comprehensive experiments.

- We validate the effectiveness of the proposed method through extensive experiments on six bench-mark datasets, demonstrating consistent performance gains across multiple SNN architectures. Compared to both the baseline and a naive temporal extension variant and channel axis concatenation, our method significantly reduces RRSE—achieving up to 6.5% reduction—highlighting the practical advantage of integrating channel similarity into spike-form encoding.

## 2  RELATED WORKS

This section provides an overview of prior studies on two key directions relevant to our work: (1) time-series forecasting approaches that leverage inter-channel similarity in multivariate data and (2) recent developments in applying Spiking Neural Networks (SNNs) to multivariate time-series tasks.

### 2.1  TIME-SERIES FORECASTING WITH CHANNEL INFORMATION

Recent work has emphasized the importance of explicitly modeling inter-channel dependencies for multivariate time-series forecasting. DUET (Qiu et al., 2025) combines temporal and channel clustering to capture heterogeneity and suppress noise, while a unified CCM framework (Chen et al., 2024a) dynamically groups channels via prototype embeddings to balance CI and CD approaches. In addition, LIFT (Zhao & Shen, 2024) leverages lead–lag relationships through cross-correlation, yielding consistent improvements. Collectively, these studies demonstrate that channel interaction modeling leads to substantial gains in forecasting accuracy.

### 2.2  SNN-BASED MULTIVARIATE TIME-SERIES PROCESSING

Recent advances in SNN forecasting have focused on improving spike and temporal encoding as well as neuron design. Banerjee et al. (2022) optimized spike encoding via mutual information maximization, Lv et al. (2024b) introduced Delta and Convolutional Spike encoders for efficient spike conversion, and Lv et al. (2024a; 2025) proposed biologically inspired positional encodings based on CPG dynamics, Gray code, and logarithmic schemes. In addition, Feng et al. (2025) developed the TS-LIF neuron to enhance expressiveness by capturing both short- and long-term dependencies. While these methods have yielded significant improvements in performance and efficiency, they primarily address intra-temporal dynamics or neuron-level encoding. Despite the proven benefits of inter-channel modeling in ANN-based forecasting, SNNs still lack a spike-compatible strategy for explicitly incorporating inter-channel dependencies. Our work fills this gap by quantifying such dependencies and integrating them into SNN forecasting models.

## 3 PRELIMINARY

### 3.1 TIME SERIES FORECASTING

Time-series forecasting is the task of predicting future observations from past data. Given a multivariate time series input $X = [x_1, x_2, \ldots, x_T] \in \mathbb{R}^{T \times C}$, where each $x_t \in \mathbb{R}^C$ represents the observation across $C$ channels at time step $t$, we use $X_{[:,i]} \in \mathbb{R}^T$ (denoted as $X_i$ for simplicity) to represent the temporal sequence of the $i$-th channel. The goal is to estimate the future sequence

$$Y = [y_{T+1}, y_{T+2}, \ldots, y_{T+H}] \in \mathbb{R}^{H \times C}, \tag{1}$$

where $H$ denotes the prediction horizon. Formally, we learn parametric mapping

$$\hat{Y} = f_\theta(X), \tag{2}$$

with learnable parameters $\theta$. The function $f_\theta$ is typically implemented using neural architectures that can capture temporal dependencies, inter-variable interactions, and dynamic patterns.

### 3.2 SPIKING NEURON AND SURROGATE GRADIENT

The leaky integrate-and-fire (LIF) neuron accumulates incoming current over time, emits a spike when the membrane potential exceeds a threshold, and resets its potential afterward. In discrete time, the membrane potential update is defined as

$$u[t + 1] = \alpha u[t](1 - s[t]) + I[t], \tag{3}$$

where $u[t]$ is the membrane potential at time $t$, $\alpha \in (0, 1)$ is the decay factor, $I[t]$ the input current, and $s[t]$ is a binary indicator of spike firing obtained by

$$s[t] = H(u[t] - V_{th}). \tag{4}$$

Here, $H$ denotes the Heaviside function, and $V_{th}$ is the firing threshold that determines whether the neuron emits a spike. When a spike occurs, the potential is reset, which is modeled via the $(1 - s[t])$ term. Because this binary firing operation is non-differentiable, we adopt a surrogate gradient to approximate $s[t]$ with a smooth function, allowing the model to be trained using backpropagation through time(BPTT).

### 3.3 TEMPORAL ALIGNMENT AND CONVOLUTIONAL SPIKE ENCODER

To effectively adapt time-series data to the SNN framework, it is essential to align the time unit of the original sequence $\Delta T$ with the fine-grained time step of the SNN $\Delta t$. Following prior work (Lv et al., 2024b), we divide each $\Delta T$ interval into $T_s$ discrete steps of length $\Delta t$, such that $\Delta T = T_s \Delta t$. Within each interval, the membrane potential $u[t]$ evolves according to Equation 3, and a spike may be emitted according to the firing condition in Equation 4. This alignment ensures temporal synchronization between the input sequence $X$ and the internal dynamics of the SNN, including the membrane potential $u[t]$, the input current $I[t]$, the spike output $s[t]$, and the Heaviside activation function $H$. This temporal structure forms the foundation for designing effective spike encoding mechanisms that map continuous-valued inputs into spike trains compatible with the SNN's time resolution. We adopted the Convolutional Spike Encoder as the encoder for transforming input time-series into spike representations. This encoder is designed to effectively capture the morphological information of time-series data, by passing the input sequence $X \in \mathbb{R}^{T \times C}$ through a convolutional layer and batch normalization, followed by spiking neurons that emit spikes when the subsequence shape matches the kernel. This process can be expressed as

$$S = SN(BN(Conv(X))). \tag{5}$$

As a result, the spike train is expanded to $T_s \times T \times C$, reflecting the matching outcomes with various kernels at each time step.

## 4 METHODOLOGY

### 4.1 METHOD OVERVIEW

Figure 1 illustrates the proposed channel-similarity encoding(CSE) module, which can be integrated into existing SNN-based forecasting models. The module first computes soft cluster assignments for

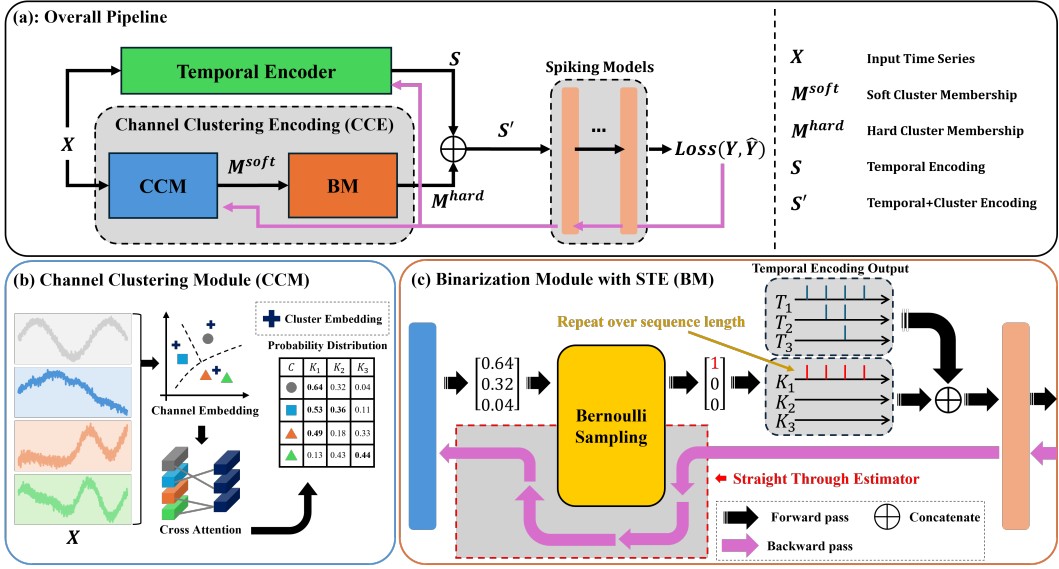

Figure 1: Illustration of the proposed Channel-Similarity Encoding (CSE) module. (a) Overall pipeline integrating the CSE module into existing SNN-based forecasting architectures. (b) Channel Clustering Module (CCM): each input channel is projected to a shared embedding space and assigned to soft clusters via attention-based similarity. (c) Spike-form Binarization Module (BM) with Straight Through Estimator (STE): the soft assignments are binarized using Bernoulli sampling with a STE, and the resulting binary cluster indicators are concatenated along the SNN time axis of the temporal encoding.

each channel via attention-based clustering, then converts them into binary spike representations using a Straight-Through Estimator(STE) (Bengio et al., 2013). These spike-form cluster indicators are concatenated along the spike time dimension of the temporal encoding output, enabling subsequent layers to leverage spike encodings that integrate both temporal dynamics and channel structural information. Each step of the module is described in detail in the following subsections. For clarity, a complete summary of the notation used in this module is provided in Appendix A.2.

## 4.2 SIMILARITY-BASED CHANNEL CLUSTERING

The overall procedure is illustrated in Figure 1-(b), which shows how the input channels are projected into a shared embedding space and assigned to soft clusters using attention-based similarity. To quantitatively capture inter-channel information in multivariate time-series data, we adopt the attention-based clustering framework (Chen et al., 2024a). This module probabilistically estimates a cluster membership for each channel and is used to model the properties of every cluster. We initialize $K$ learnable cluster embeddings

$$C = \{c_1, c_2, \ldots, c_K\}, \text{ where } c_k \in \mathbb{R}^d. \tag{6}$$

Each channel $X_i \in \mathbb{R}^T$ of the input time series $X$ is mapped to a $d$-dimensional embedding $h_i \in \mathbb{R}^d$ through a multilayer perceptron, where $i$ denotes the channel index ($i = 1, \ldots, C$). The probability that channel $X_i$ belongs to $k$-th cluster, denoted as $M_{i,k}^{soft}$, is obtained by normalizing the cosine similarity

$$M_{i,k}^{soft} = \text{Normalize}\left(\frac{c_k^\top h_i}{|c_k||h_i|}\right) \in [0, 1]. \tag{7}$$

The normalization operation ensures that $\sum_k M_{i,k}^{soft} = 1$ for each $i$-th channel, thereby forming a valid probability distribution. To improve clustering quality, we incorporate a regularization loss proposed in previous work (Chen et al., 2024a), which encourages similar channels to be assigned to the same cluster while separating dissimilar channels into different clusters. Specifically, it minimizes the distance between each channel embedding and its assigned cluster center, while maximizing the distance between different cluster centers. Optimized jointly with the forecasting loss,

the regularization term is critical for producing stable soft clustering assignments before spike-form binarization.

### 4.3 SPIKE-FORM BINARIZATION

---

**Algorithm 1** SNN Time Axis Concatenation

---

1: Input: Soft Cluster Probability $M^{soft}$, Temporal Spike Encoding $S$
2: Output: $S' \in \mathbb{R}^{(T_s+K) \times L \times C}$
3: $M^{hard} \leftarrow \text{Bernoulli}(M^{soft})$
4: $M^{hard} \leftarrow M^{soft} + (M^{hard} - M^{soft}).\text{detach}()$
5: $M^{hard} \leftarrow \text{ExpandToSNNTime}(M^{hard})$
6: $S' \leftarrow \text{Concatenate}(S, M^{hard})$

---

Soft cluster assignment generated by Gumbel-Softmax (Jang et al., 2017) or Concrete Bernoulli does not satisfy the strict binary nature required for spike-based processing. We therefore employ Bernoulli sampling combined with a Straight-Through Estimator(STE) to obtain exact binary spike outputs, as illustrated in Figure 1-(c). Given the cluster probabilities $M_{i,k}^{soft} \in [0, 1]$, we first sample Then we apply STE to retain gradients Here, the binary mask $M^{hard} \in (0, 1)^{K \times C}$, labeled as Hard Cluster Membership in Figure 1, is used during forward pass, while the gradient flows through the continuous value $M_{i,k}^{soft}$-denoted as $M^{soft}$ in Figure 1-in the backward pass. This design produces spike-compatible signals while supporting gradient-based optimization.

### 4.4 CLUSTER INFORMATION INTEGRATION

The binary spike-form cluster membership $M^{hard} \in \{0, 1\}^{K \times C}$ encodes the cluster affiliation of each channel and is injected into the forecasting backbone. Specifically, we propose integrating this information into temporal encoding output $S \in \mathbb{R}^{T_s \times T \times C}$, where $T_s$ is the number of SNN time steps and $T$ is the time-series dimension. The key idea is to concatenate the cluster representation along the SNN time axis $T_s$, enabling the model to integrate both temporal dynamics and inter-channel structural information. We replicate the cluster membership matrix $M^{hard}$ along $T$ axes, forming a tensor of shape $K \times T \times C$ and concatenate it with $S$ along the SNN time axis by extending the temporal resolution from $T_s$ to $T_s + K$, the model can process both temporal dynamics $T_s$ and inter-channel structural information $K$ as a part of the input representation. This design uses concatenation rather than addition to preserve binary spike values and prevent unintended floating-point operations (Lv et al., 2024a).

## 5 EXPERIMENTS

### 5.1 EXPERIMENTAL SETUP

We evaluate our method on six widely used multivariate time series benchmarks: Electricity (Trindade, 2015), Solar (Lai et al., 2018), ETTh1 (Zhou et al., 2021a), ETTh2 (Zhou et al., 2021a), METR-LA (Li et al., 2017), and Weather (Wu et al., 2021). For each dataset, forecasting horizons are set to 6, 24, 48, and 96 steps ahead, following standard practice. We compare our proposed strategy against two alternative approaches—SNN Time extension and Channel concatenation—across six representative architectures: RNN (Rumelhart et al., 1985), Spike-RNN (Lv et al., 2024b), iTransformer (Liu et al., 2023), Spikformer (Zhou et al., 2022), TCN (Bai, 2018), and Spike-TCN (Lv et al., 2024b). All models are trained using the Adam optimizer with a learning rate of 1e-3. Training is conducted for up to 1,000 epochs, with early stopping applied if no improvement is observed over 30 consecutive epochs. The mean squared error (MSE) is used as the training loss function, while evaluation is performed with root relative squared error (RRSE) to ensure comparability across datasets of different scales. Our implementation is based on PyTorch, leveraging SpikingJelly and snnTorch for spiking neural network modules. Experiments are executed on a server running Ubuntu 20.04 with four NVIDIA RTX A4000 GPUs.

| Dataset | Architecture | RNN | Spike-RNN | | iTrans. | Spikformer | | TCN | Spike-TCN | |
|---|---|---|---|---|---|---|---|---|---|---|
| | **Method** | - | Baseline | Ours | - | Baseline | Ours | - | Baseline | Ours |
| Electricity | 6 | 0.239 | 0.490 | 0.377 | 0.139 | 0.366 | 0.332 | 0.258 | 0.331 | 0.351 |
| | 24 | 0.284 | 0.478 | 0.381 | 0.160 | 0.399 | 0.372 | 0.580 | 0.348 | 0.365 |
| | 48 | 0.293 | 0.578 | 0.409 | 0.189 | 0.363 | 0.393 | 0.382 | 0.378 | 0.359 |
| | 96 | 0.341 | 0.493 | 0.392 | 0.226 | 0.409 | 0.381 | 0.400 | 0.389 | 0.350 |
| Solar | 6 | 0.272 | 0.287 | 0.251 | 0.202 | 0.480 | 0.439 | 0.196 | 0.238 | 0.234 |
| | 24 | 0.404 | 0.416 | 0.369 | 0.355 | 0.651 | 0.550 | 0.464 | 0.390 | 0.382 |
| | 48 | 0.528 | 0.510 | 0.505 | 0.452 | 0.794 | 0.744 | 0.550 | 0.552 | 0.546 |
| | 96 | 0.607 | 0.552 | 0.531 | 0.514 | 0.888 | 0.825 | 0.619 | 0.706 | 0.684 |
| ETTh1 | 6 | 0.556 | 0.743 | 0.665 | 0.475 | 1.028 | 1.022 | 0.654 | 0.641 | 0.626 |
| | 24 | 0.678 | 0.842 | 0.830 | 0.562 | 1.029 | 1.052 | 0.862 | 0.840 | 0.837 |
| | 48 | 0.736 | 0.795 | 0.849 | 0.609 | 1.050 | 1.031 | 0.850 | 0.824 | 0.824 |
| | 96 | 0.788 | 0.943 | 0.878 | 0.652 | 1.039 | 1.023 | 0.935 | 0.831 | 0.829 |
| ETTh2 | 6 | 0.689 | 0.801 | 0.796 | 0.433 | 1.582 | 1.448 | 0.722 | 1.455 | 1.387 |
| | 24 | 0.791 | 0.870 | 0.904 | 0.540 | 1.458 | 1.453 | 1.225 | 1.593 | 1.524 |
| | 48 | 0.933 | 1.044 | 0.918 | 0.603 | 1.568 | 1.381 | 1.237 | 1.817 | 1.552 |
| | 96 | 1.073 | 1.072 | 0.987 | 0.675 | 1.505 | 1.346 | 1.276 | 1.681 | 1.688 |
| METR-LA | 6 | 0.575 | 0.558 | 0.555 | 0.408 | 0.665 | 0.605 | 0.460 | 0.478 | 0.476 |
| | 24 | 0.699 | 0.703 | 0.702 | 0.652 | 0.763 | 0.762 | 0.670 | 0.668 | 0.668 |
| | 48 | 0.799 | 0.794 | 0.790 | 0.808 | 0.833 | 0.804 | 0.779 | 0.780 | 0.780 |
| | 96 | 0.904 | 0.889 | 0.868 | 0.936 | 0.906 | 0.913 | 0.895 | 0.872 | 0.865 |
| Weather | 6 | 0.628 | 0.584 | 0.569 | 0.457 | 0.965 | 0.872 | 0.453 | 0.743 | 0.712 |
| | 24 | 0.642 | 0.730 | 0.688 | 0.573 | 1.080 | 1.002 | 0.565 | 0.825 | 0.824 |
| | 48 | 0.745 | 0.790 | 0.784 | 0.651 | 1.272 | 1.051 | 0.656 | 0.890 | 0.883 |
| | 96 | 1.244 | 0.865 | 0.848 | 0.730 | 1.322 | 1.155 | 0.774 | 1.072 | 1.014 |
| **Average** | | 0.644 | 0.701 | 0.660 | 0.500 | 0.934 | 0.873 | 0.686 | 0.806 | 0.782 |

Abbreviation: iTrans. = iTransformer

Table 1: Main Result

## 5.2 MAIN RESULT

As shown in Table 1, the proposed method improves performance across most spike-based architectures and datasets. On average, Spike-RNN, Spikformer, and Spike-TCN achieved relative error reductions of 5.8%, 6.5%, and 3.0%, respectively, compared to their baselines, amounting to an overall 5.2% improvement. Dataset-wise analysis reveals varying degrees of effectiveness. On Electricity, Spike-RNN and Spikformer exhibit notable improvements, while Spike-TCN shows mixed results with slight degradations at shorter horizons. On Solar, our method yields improvements across all spike models. On ETTh1, Spike-RNN and Spike-TCN benefit clearly, whereas Spikformer gains are marginal or near-identical to the baseline. In ETTh2, Spikformer and Spike-TCN achieve substantial reductions in error (up to 14.6%), and Spike-RNN also improves slightly. On METR-LA, Spikformer shows clear improvements, while Spike-RNN and Spike-TCN yield near-identical results to their baselines. Finally, on Weather, our method enhances all spike models, with Spikformer achieving the largest reduction (-12.6% at horizon 96). Figure 2 visualizes the relative performance gaps between ANN baselines, SNN baselines, and the proposed method across all datasets, illustrating how the proposed approach consistently reduces the performance gap between SNNs and their ANN counterparts. Across datasets, errors naturally increase with longer prediction horizons, yet the proposed method remains effective, particularly on ETTh2 and Weather, demonstrating robustness under more challenging settings. Although a few instances show marginal gains or slight regressions, such as in Spike-TCN on Electricity and Spikformer on ETTh1, the overall trend indicates that the proposed method provides stable benefits across spike-based architectures, with the most pronounced improvements observed for Spikformer and Spike-RNN.

## 5.3 VALIDATION AGAINST ALTERNATIVES

To validate the effectiveness of the proposed approach, we compare it against two alternative strategies for integrating cluster membership information into SNN-based forecasting models. The first alternative, SNN Time extension, expands the SNN time resolution by concatenating the binary cluster membership $M^{hard} \in \{0, 1\}^{K \times C}$ along the SNN time axis, extending $T_s$ to $T_s + K$. This approach increases the temporal resolution without directly encoding channel similarity into the spike representation. The second alternative, Channel axis concatenation, appends the cluster membership along the channel dimension, expanding the input from $C$ to $C + K$ channels. While this

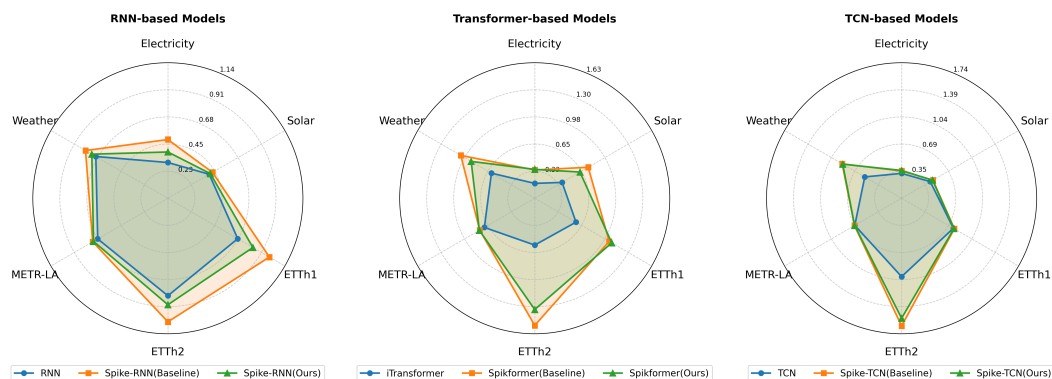

Figure 2: Radar chart comparing the prediction performance of ANN, SNN (Baseline), and SNN (Ours) across six datasets for each architecture (RNN–Spike-RNN, iTransformer–Spikformer, TCN–Spike-TCN). Since RRSE values indicate better performance when lower, proximity to the center signifies higher accuracy. This visualization was used not to compare different architectures, but to visually confirm the performance difference between ANN and spike-based structures within the same architecture, and to what extent the proposed technique improves upon SNN (Baseline).

preserves the original SNN time axis, it increases the channel-wise computation and may not fully exploit the spike-based encoding structure. In contrast, our proposed method integrates cluster membership by replicating $M^{hard}$ across the time-series dimension and concatenating it along the SNN time axis, thereby embedding channel similarity information into the spike-form temporal encoding $S' \in \mathbb{R}^{(T_s+K) \times T \times C}$. This design enables the model to leverage both temporal dynamics and inter-channel structural information within a unified spike-based representation. As shown in Table 2, across all datasets and horizons, the proposed method demonstrates improvements over the two alternative strategies. On average, Ours achieves the lowest error (0.660), outperforming both Channel concatenation (0.691) and SNN Time extension (0.786). The gains are especially clear on Electricity and Weather, where Ours outperforms the best baseline by large margins (e.g., 0.377 vs. 0.462/0.557 on Electricity at horizon 6, and 0.569 vs. 0.839/0.671 on Weather at horizon 6). On Solar, the improvements are moderate but stable, with Ours yielding the lowest error in three out of four horizons. For ETTh1 and ETTh2, Ours consistently ranks between the two baselines: it improves substantially over SNN Time extension but remains slightly higher than Channel concatenation in certain cases. In METR-LA, all three methods perform similarly, with Ours closely matching Channel concatenation. Overall, these results indicate that while Channel concatenation already provides notable benefits over SNN Time extension, the proposed method consistently pushes performance further, demonstrating robustness across diverse datasets and horizons. The largest relative improvements occur in settings where capturing inter-channel dependencies is critical (e.g., Weather and Electricity), suggesting that the proposed strategy better exploits both temporal and inter-channel information.

## 5.4 PARAMETER ANALYSIS

We further conduct a sensitivity analysis on three key hyperparameters: the number of clusters $n_{cluster}$, the hidden dimension $d_{model}$, and the regularization coefficient $\beta$. As shown in Table 3 and Figure 3, the choice of hyperparameters significantly affects forecasting performance. For the cluster size, the case of $n_{cluster} = 1$ represents a no-clustering setting where all channels are assigned to a single cluster. While this configuration does not provide inter-channel similarity information, it differs from the baseline in that a binary spike-form indicator (denoting membership to the single cluster) is still concatenated along the SNN time axis. This time-axis augmentation expands the temporal representation capacity of the SNN, which can lead to modest performance improvements over the baseline in certain datasets. However, performance improves more markedly when increasing $n_{cluster}$ from 1 to 3, with $n_{cluster} = 3$ yielding the lowest errors across all horizons (e.g., 0.377 at horizon 6 and 0.409 at horizon 48). This demonstrates that the incorporation of meaningful channel-similarity information through clustering provides additional benefits beyond the simple expansion

| Dataset | Horizon | Strategy | | |
|---|---|---|---|---|
| | | SNN Time↑ | Channel | Ours |
| Electricity | 6 | 0.462 | 0.557 | 0.377 |
| | 24 | 0.475 | 0.520 | 0.381 |
| | 48 | 0.489 | 0.489 | 0.409 |
| | 96 | 0.481 | 0.433 | 0.392 |
| Solar | 6 | 0.262 | 0.267 | 0.251 |
| | 24 | 0.430 | 0.400 | 0.369 |
| | 48 | 0.512 | 0.471 | 0.505 |
| | 96 | 0.563 | 0.510 | 0.531 |
| ETTh1 | 6 | 0.771 | 0.570 | 0.665 |
| | 24 | 0.843 | 0.795 | 0.830 |
| | 48 | 0.865 | 0.844 | 0.849 |
| | 96 | 0.955 | 0.812 | 0.878 |
| ETTh2 | 6 | 1.076 | 0.682 | 0.796 |
| | 24 | 1.200 | 1.021 | 0.904 |
| | 48 | 1.257 | 0.990 | 0.918 |
| | 96 | 1.293 | 1.237 | 0.987 |
| METR-LA | 6 | 0.561 | 0.560 | 0.555 |
| | 24 | 0.709 | 0.702 | 0.702 |
| | 48 | 0.792 | 0.786 | 0.790 |
| | 96 | 0.871 | 0.881 | 0.868 |
| Weather | 6 | 0.839 | 0.671 | 0.569 |
| | 24 | 0.949 | 0.720 | 0.688 |
| | 48 | 0.996 | 0.783 | 0.784 |
| | 96 | 1.206 | 0.879 | 0.848 |
| **Average** | | 0.786 | 0.691 | 0.660 |

**SNN Time**↑: Extends SNN time axis from $T_s$ to $T_s + K$ by concatenating cluster membership along the time dimension.
**Channel**: Expands input channels from $C$ to $C + K$ by concatenating cluster membership along the channel dimension.
**Ours**: Replicates cluster membership across time-series dimension and concatenates along SNN time axis, resulting in $S' \in \mathbb{R}^{(T_s+K) \times T \times C}$.

Table 2: Validation against Alternative Approaches

of the temporal axis. However, further increasing $n_{cluster}$ to 4 results in performance degradation. This decline can be attributed to over-partitioning: when $n_{cluster} \geq 4$, empty clusters (clusters to which no channels are assigned) emerge on the Electricity dataset, and similar tendencies are observed on other datasets when $n_{cluster}$ exceeds 3 or 4. Such over-partitioning destabilizes channel clustering and dilutes the structural information conveyed through spike-form indicators, as visually evident in Figure 4. Rather than suggesting an intrinsic "optimal number of groups" inherent to the data, this finding indicates that exceeding a certain cluster threshold degrades the stability of the spike-form integration process, leading to reduced forecasting performance. Additional visualizations of the learned channel clusters are provided in Appendix A.3, confirming that channels with similar temporal patterns tend to be grouped together when $n_{cluster}$ is appropriately chosen.

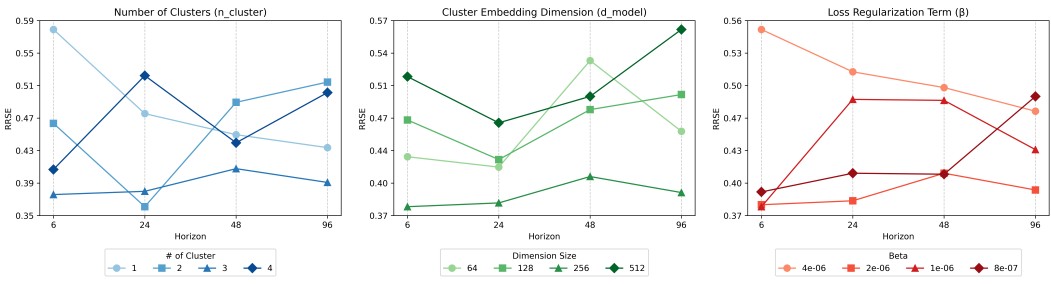

Figure 3: Parameter Analysis

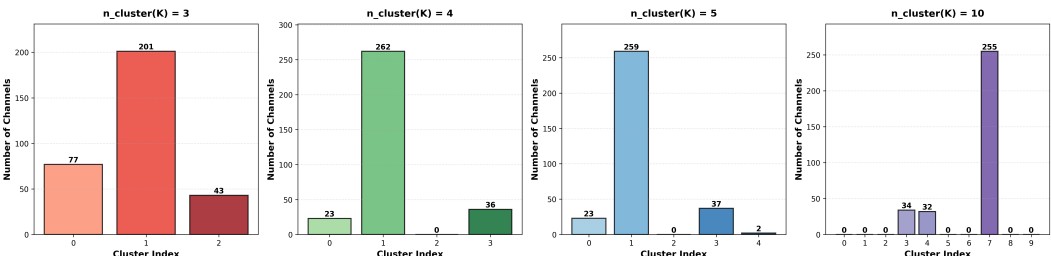

Figure 4: Visualization of Cluster Membership for Different Number of Clusters on Electricity Dataset. Each bar represents the number of channels, When $n_{cluster} \geq 4$, some clusters remain unassigned.

| Dataset | | Horizon | | | |
|---|---|---|---|---|---|
| | | 6 | 24 | 48 | 96 |
| Baseline | | 0.490 | 0.478 | 0.578 | 0.493 |
| n_cluster | 1 | 0.581 | 0.477 | 0.451 | 0.435 |
| | 2 | 0.465 | 0.362 | 0.491 | 0.516 |
| | 3 | 0.377 | 0.381 | 0.409 | 0.392 |
| | 4 | 0.408 | 0.524 | 0.441 | 0.503 |
| d_model | 64 | 0.430 | 0.419 | 0.532 | 0.457 |
| | 128 | 0.469 | 0.427 | 0.480 | 0.496 |
| | 256 | 0.377 | 0.381 | 0.409 | 0.392 |
| | 512 | 0.515 | 0.466 | 0.494 | 0.565 |
| $\beta$ | 4e-6 | 0.555 | 0.512 | 0.496 | 0.472 |
| | 2e-6 | 0.377 | 0.381 | 0.409 | 0.392 |
| | 1e-6 | 0.375 | 0.484 | 0.483 | 0.433 |
| | 8e-7 | 0.39 | 0.409 | 0.408 | 0.487 |

Table 3: Parameter Analysis

Similarly, the hidden dimension of $d_{model} = 256$ achieves consistently superior results compared to both smaller (64, 128) and larger (512) dimensions, suggesting a balance between model capacity and overfitting. Regarding the coefficient $\beta$, the setting of $2 \times 10^{-6}$ provides the most stable improvements, while larger or smaller values lead to performance degradation. Overall, the sensitivity study indicates that the configuration of $n_{cluster} = 3$, $d_{model} = 256$, and $\beta = 2 \times 10^{-6}$ offers the most reliable performance across different forecasting horizons.

## 5.5 THEORETICAL ENERGY CONSUMPTION

Table 4 reports the theoretical energy consumption estimated on the Electricity dataset using an input sequence length of 168 and a forecasting horizon of 96. Under this setting, the number of clusters was fixed to $K = 3$, which results in extending the SNN time axis from the baseline value of $T_s = 4$ to $T_s = 4 + 3$. As shown in Table 4, applying the proposed method increases the theoretical energy consumption by $3.3\times$ in Spike-RNN and $2.7\times$ in Spike-TCN compared with their baselines. This increase arises from three factors. First, the introduction of the clustering module slightly increases $FLOPs$ (e.g., $+6.2\%$ in Spike-RNN and $+9.3\%$ in Spike-TCN), although this alone accounts for

| Architecture | Method | $FLOPs$ | $\gamma$ | $T_s$ | Energy (mJ) | Reduction |
|---|---|---|---|---|---|---|
| RNN | - | 27.6M | - | - | 0.127 | - |
| Spike-RNN | Baseline | 80.4M | 3.2% | 4 | 0.009 | 92.7% ↓ |
| | Ours | 85.4M | 5.7% | 4+3 | 0.030 | 75.9% ↓ |
| TCN | - | 13.3G | - | - | 6.105 | - |
| Spike-TCN | Baseline | 24.8G | 9.5% | 4 | 0.852 | 86.1% ↓ |
| | Ours | 27.1G | 13.3% | 4+3 | 2.275 | 62.7% ↓ |

Table 4: Theoretical Energy Consumption Estimation

only a small portion of the overall growth. Second, concatenating the binary cluster indicators to the temporal spike encoding enlarges the effective input size fed into subsequent SNN layers, thereby increasing the number of synaptic events. Third, extending the SNN time axis from 4 to $4 + 3$ directly raises the number of membrane updates and spike-generation opportunities. This effect is further accompanied by the increase in the firing rate $\gamma$ (e.g., $3.2\% \rightarrow 5.7\%$ in Spike-RNN), which contributes noticeably to total energy usage given the dominance of synaptic operations in SNNs. These factors jointly explain why the energy increase is greater than what the $FLOPs$ increase alone would suggest. We note that similar trends were observed empirically across the other datasets used in this study. Although this overhead is presented, the proposed method still maintains substantial energy advantages over ANN baselines (e.g., $75.9\%$ reduction for RNN and $62.7\%$ for TCN) while providing consistent forecasting improvements across multiple spike-based architectures. These results suggest that incorporating channel similarity into spike-form encoding enhances representational capacity with moderate additional cost, a tendency that aligns with the broader observations discussed in Section 6. The detailed estimation procedure and the underlying energy model are provided in Appendix A.1. Table 4 summarizes the $FLOPs$, firing rate $\gamma$, SNN time steps $T_s$, and the resulting theoretical energy consumption for clarity.

## 6 DISCUSSION

This work demonstrates that integrating inter-channel similarity information into spike-based encoding enhances SNN forecasting performance while maintaining computational efficiency. Experimental results show an average RRSE reduction of 5.2% across spike-based architectures, with particularly strong improvements on datasets where inter-channel dependencies are salient (e.g., Electricity, ETTh2, Weather). However, the proposed method incurs a theoretical energy increase of approximately 2.7–3.3 times compared to SNN baselines, primarily due to extending the SNN time axis from $T_s$ to $T_s + K$, which increases membrane updates, spike-generation opportunities, and firing rates. Despite this overhead, the method still maintains lower energy consumption than ANN baselines (75.9% reduction for RNN, 62.7% for TCN), positioning it as a viable option where moderate energy–accuracy trade-offs are acceptable, though the roughly 3-fold energy increase limits applicability in extreme energy-constrained environments. Several limitations warrant acknowledgment. First, while time-axis concatenation preserves spike-form representation, alternative integration methods such as threshold modulation or lightweight auxiliary networks could potentially achieve similar benefits with lower energy overhead. Second, the static clustering approach does not account for time-varying channel relationships, and future work could explore dynamic clustering mechanisms. Third, the performance degradation when $n_{cluster} \geq 4$ stems from over-partitioning that creates empty clusters and dilutes structural information. Finally, while our method improves SNN performance, a gap remains compared to state-of-the-art ANNs, reflecting fundamental structural differences between dense and event-driven computation. Future research directions include exploring more energy-efficient integration strategies, investigating dynamic clustering mechanisms, and validating the approach on actual neuromorphic hardware platforms.

## 7 CONCLUSION

This paper proposed a spike-compatible channel-similarity encoding framework for multivariate time-series forecasting with SNNs. We introduced a method that quantifies inter-channel similarity through attention-based clustering and integrates it into spike-based temporal encoding via a Straight-Through Estimator (STE), enabling SNNs to leverage both temporal dynamics and channel structural information. Experimental evaluation on six benchmark datasets demonstrated average RRSE reductions of 5.2%, with the proposed approach achieving the lowest average error (0.660 vs. 0.691 and 0.786) compared to alternative integration strategies. However, extending the SNN time axis increases theoretical energy consumption by approximately 2.7–3.3 times compared to SNN baselines, though the method still maintains energy advantages over ANN baselines. This work demonstrates that inter-channel similarity modeling represents a promising dimension for advancing SNN-based forecasting, with future directions including exploring more energy-efficient integration strategies and validating the approach on neuromorphic hardware platforms.

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

# A  APPENDIX

## A.1  THEORETICAL ENERGY CONSUMPTION ESTIMATION

This appendix describes the procedure used to compute the theoretical energy consumption values reported in Table 4. The estimation follows widely used analytical models in neuromorphic computing and is intended to provide a hardware-agnostic comparison of relative energy trends across architectures. As with prior work, the values reported here should be interpreted as theoretical approximations rather than exact measurements of any specific hardware platform. We adopt the 45nm neuromorphic hardware assumption commonly used in the literature (Yao et al., 2023; Lv et al., 2024b), where the energy cost of a multiply–accumulate (MAC) operation and an accumulate (AC) operation are given by $E_{\text{MAC}} = 4.6$ pJ and $E_{\text{AC}} = 0.9$ pJ, respectively. Because AC operations omit the multiplication term required in MAC operations, they consume substantially less energy, which explains why SNNs generally achieve lower per-operation energy costs compared with ANNs. Under this setting, the theoretical energy consumption of ANN and SNN models is computed as

$$E_{\text{ANN}} = FLOPs \times E_{\text{MAC}}, \tag{8}$$

$$E_{\text{SNN}} = E_{\text{AC}} \times SOPs. \tag{9}$$

$$SOPs = FLOPs \times \gamma \times T_s, \tag{10}$$

For SNNs, the number of synaptic operations ($SOPs$) is determined by the amount of inherited floating-point computation, the firing rate $\gamma$, and the total number of SNN time steps $T_s$. Based on related studies (Guo et al., 2024), the firing rate $\gamma$ was computed empirically by measuring the proportion of active spikes over all neurons and all SNN time steps during a forward pass, i.e., the ratio of nonzero spike events to the maximum possible number of spike activations. Using these equations and measured quantities ($FLOPs$, firing rate $\gamma$, and $T_s$), the theoretical energy consumption values for both ANN and SNN architectures in Table 4 were obtained. These values serve as indicators of relative energy scaling rather than precise hardware-level measurements, as device-specific factors such as memory access cost or routing overhead are not modeled here.

## A.2  NOTATION

The main symbols used in this paper are summarized in Table 5. The notation presented in the table aims to consistently represent key elements recurring throughout this study, ranging from the

basic structure of the input time series $(X, Y, T, C, H)$ to the embedding vectors $(c_k, h_i)$ and cluster assignment matrices $(M^{soft}, M^{hard})$, and the variables defining the spike-based representation $(S, T_s, \gamma)$. Furthermore, the regularization coefficient $\beta$ and input/output-related variables are essential for understanding the learning process and performance analysis of the proposed technique. This table is provided to help readers interpret the equations and algorithms in the main text clearly and efficiently.

| Notation | Description |
|---|---|
| $X \in \mathbb{R}^{T \times C}$ | Input multivariate time series |
| $T$ | Length of input sequence |
| $C$ | Number of channels |
| $H$ | Prediction horizon |
| $Y \in \mathbb{R}^{H \times C}$ | Target future sequence |
| $K$ | Number of clusters |
| $d$ | Embedding dimension |
| $c_k \in \mathbb{R}^d$ | $k$-th cluster embedding |
| $h_i \in \mathbb{R}^d$ | Embedding of $i$-th channel |
| $M^{soft} \in [0,1]^{C \times K}$ | Soft cluster assignment probabilities |
| $M^{hard} \in \{0,1\}^{C \times K}$ | Binary cluster membership |
| $S \in \mathbb{R}^{T_s \times T \times C}$ | Temporal spike encoding |
| $S' \in \mathbb{R}^{(T_s+K) \times T \times C}$ | Channel-similarity aware spike encoding |
| $T_s$ | Number of SNN time steps |
| $u[t]$ | Membrane potential at time $t$ |
| $s[t]$ | Spike output at time $t$ |
| $\alpha$ | Membrane decay factor |
| $V_{th}$ | Firing threshold |
| $\gamma$ | Firing rate |
| $\beta$ | Regularization coefficient |

Table 5: Summary of Notation

## A.3 VISUALIZATION OF LEARNED CHANNEL CLUSTERS

This appendix provides additional visualizations of the learned channel clusters on the Electricity dataset when $n_{cluster} = 3$. Figures 5, 6, and 7 illustrate representative channels from Cluster 1, Cluster 2, and Cluster 3, respectively. Due to the large number of channels in the Electricity dataset (321 channels), it is not feasible to visualize all channels simultaneously. Instead, we present a subset of representative channels from each cluster to demonstrate that channels with similar temporal patterns tend to be grouped together when $n_{cluster}$ is appropriately chosen. As shown in the figures, channels within the same cluster exhibit similar temporal dynamics, confirming that the attention-based clustering module successfully captures inter-channel similarity.

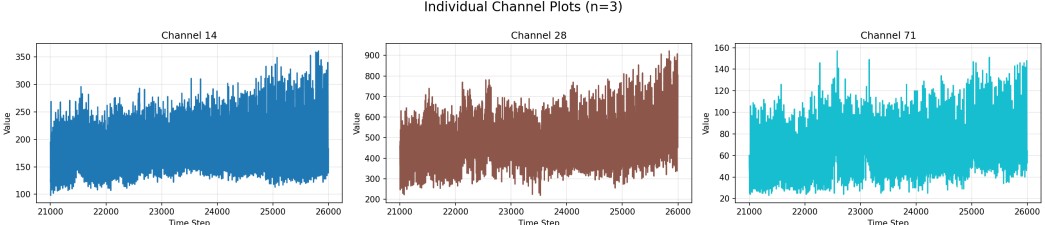

Figure 5: Representative channels from Cluster 1 when $n_{cluster} = 3$

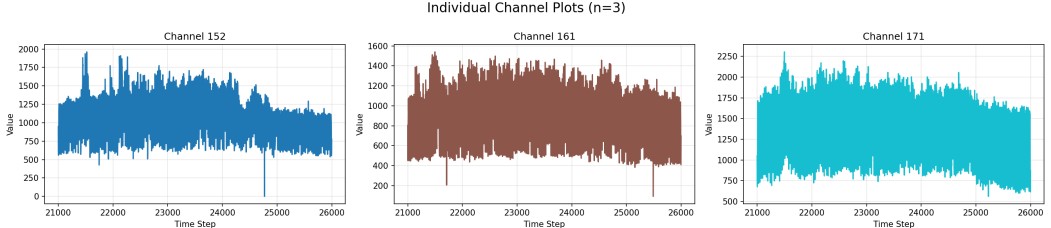

Figure 6: Representative channels from Cluster 2 when $n_{cluster} = 3$

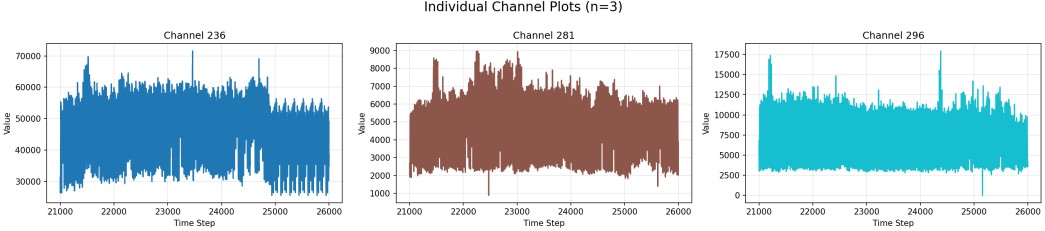

Figure 7: Representative channels from Cluster 3 when $n_{cluster} = 3$

