# OpenReview forum: "Channel-Similarity Aware Spike Encoding for Multivariate Time-Series Forecasting"
_ICLR.cc/2026/Conference — Submitted to ICLR 2026_

### Official Review · Reviewer_YQYf · 2025-10-26

**Soundness:** 2
**Presentation:** 1
**Contribution:** 2
**Rating:** 2
**Confidence:** 3

**Summary:**

This paper proposes a preprocessing module with channel-similarity aware spike encoding for multivariate time-series forecasting. It provides additional spikes as subsequent model's input, and is shown to be useful in time-series forecasting.

**Strengths:**

The proposed method sounds reasonable for time-series forecasting.

**Weaknesses:**

1. There are some writing problems. For example, in line 242, 'Then', 'Here' are in capital form without full stops. In line 254, 'fo' should be 'of' and 'By' should be 'by'. Table 1, 2, 3 and Figure 2, 3 are not referred in the main text. In line 268, the works regarding RNN, Spike-RNN, iTransformer, Spikformer, TCN, and Spike-TCN are not cited.
2. See questions.

**Questions:**

1. In the method part, what is the relationship of dimension variables, including $d$, $C$, $K$, $T$? What is the dimension of $X_i$ and $M$? I am not quite sure about them.
2. In Figure 2, what is the meaning of Rank? If it is the rank of RRSE or some other indicators, it's strange to put it as an axis in the figure, since the two axes has similar meanings.
3. In Table 1, why there are only 3 baseline/ours with 6 architectures? I cannot get it.
4. In Table 2, what is the setting of 'SNN Time' and 'Channel'? It seems that they are not clearly stated.
5. In Table 4, why the energy is increased so much with the proposed method? the FLOPs is not increased so much. By the way, does 'fr' mean firing rate?

---

> ### Author Response · Authors · 2025-12-04
>
> ## Weakness 1: Writing Quality Issues and Missing Citations
> We sincerely appreciate the reviewer's careful attention to the writing quality and citation completeness of our manuscript. The concerns raised regarding (i) unnecessary capitalization and missing punctuation (line 242), (ii) typo corrections ('fo' → 'of', 'By' → 'by') (line 254), (iii) omission of in-text references to Tables 1–3 and Figures 2–3, and (iv) missing citations for RNN [1], Spike-RNN [2], iTransformer [3], Spikformer [4], TCN [5], and Spike-TCN [2] (line 268) are all valid and have been carefully addressed in the current revision. We have revised the text to clearly reference all tables and figures, and added the previously omitted citations at appropriate locations to ensure a natural flow throughout the paper.
>
> [1] Rumelhart, D., Hinton, G., & Williams, R. (1985). Learning internal representations by error propagation.
>
> [2] Lv, C., Wang, Y., Han, D., Zheng, X., Huang, X., & Li, D. (2024). Efficient and effective time-series forecasting with spiking neural networks. arXiv preprint arXiv:2402.01533.
>
> [3] Liu, Y., Hu, T., Zhang, H., Wu, H., Wang, S., Ma, L., & Long, M. (2023). itransformer: Inverted transformers are effective for time series forecasting. arXiv preprint arXiv:2310.06625.
>
> [4] Zhou, Z., Zhu, Y., He, C., Wang, Y., Yan, S., Tian, Y., & Yuan, L. (2022). Spikformer: When spiking neural network meets transformer. arXiv preprint arXiv:2209.15425.
>
> [5] Bai, S. (2018). An Empirical Evaluation of Generic Convolutional and Recurrent Networks for Sequence Modeling. arXiv preprint arXiv:1803.01271.
>
> ## Question 1: Clarification of Dimension Variables and Tensor Shapes
> We greatly appreciate the reviewer's concern regarding the clarity of dimension variables and tensor shape definitions. We acknowledge that the symbols ($d$, $C$, $K$, $T$) and tensor dimension definitions in the main text could have been presented more clearly. In response to this valuable feedback, we have summarized the relationships between these variables and the dimensions of key tensors below, and incorporated these clarifications into the revision with more detailed descriptions in Sections 4.2 and 4.4.
>
> - $d$: Hidden dimensions of the channel embedding
> - $C$: Number of channels in the input multivariate time series
> - $K$: Number of clusters
> - $T$: Temporal length of the input time series (sliding window length)
> - $X_i\in \mathbb{R}^T$: Time series of the i-th channel
> - $M\in \mathbb{R}^{C×K}$: Membership indicating which cluster each channel belongs to
>
> We also thank the reviewer for noting that the $M$ in Section 4.4 referred to $M^{hard}$, but the omission of "hard" in the main text could have caused confusion. This has been corrected in the revision, and the distinction between $M^{soft}$ and $M^{hard}$ is now clearly explained in the Appendix A.2.

---

> > ### Author Response · Authors · 2025-12-04
> >
> > ## Question 2: Interpretation of "Rank" in Figure 2
> > We appreciate the reviewer's insightful observation regarding the Rank indicator in Figure 2. We acknowledge that since the Rank was directly derived from RRSE, it does not provide sufficient new information and may instead cause confusion. In response to this valuable feedback, we have completely replaced Figure 2 in the revision. We restructured the RRSE into a Radar Chart format to enable an intuitive comparison of the performance distributions of the three methods for each architecture. We believe this change not only addresses the concern about redundant axis meanings but also more clearly reveals the relative strengths and weaknesses of each method at the architecture level.
> >
> > ## Question 3: Inconsistency in the Number of Baseline/Ours Models in Table 1
> > We sincerely thank the reviewer for bringing attention to the potential confusion regarding the distinction between Tables 1 and 2. We recognize that the different experimental purposes of these two tables were not sufficiently clarified in the main text. Table 1 directly compares only the baseline spike-based model and the proposed method for each architecture (6 architectures), while Table 2 presents experimental results comparing three strategies for integrating cluster information (SNN Time Extension, Channel Concatenation, Ours) under identical conditions. Therefore, Table 1 contains only Baseline/Ours pairs and does not include all three baseline strategies, as the two tables address experiments with different objectives. We have added clearer explanations of the comparison objectives for both Tables 1 and 2.
> >
> > ## Question 4: Undefined Settings of "SNN Time" and "Channel" in Table 2
> > We deeply appreciate the reviewer's concern regarding the clarity of the 'SNN Time' and 'Channel' settings in Table 2. We acknowledge that these baseline strategies were not sufficiently defined in the main text, which could have caused confusion.
> >
> > 'SNN Time' represents the time resolution dimension used in the Spike Encoding process, serving to express continuous input values more finely in spike form. While Section 3.3 briefly mentioned that $T_s$ denotes the internal time axis within the SNN, we recognize that we failed to clearly explain what the baseline 'SNN Time' setting in Table 2 actually signifies. The baseline 'SNN Time' simply extends the existing time step $T_s$ by adding the number of clusters $K$ to $T_s + K$, increasing only the SNN time resolution independently of channel similarity information. In contrast, 'Channel' expands the input dimension from $C$ to $C + K$ by concatenating the cluster membership along the channel axis of the spike encoder.
> >
> > In response to this valuable feedback, we have added clear explanations of the two baseline definitions and their dimensional changes (SNN time-axis expansion / channel-axis expansion) in the paragraph immediately preceding Table 2, and have also added relevant content to the table notes. We believe the clarity of this section has been significantly improved thanks to the reviewer's insightful observation.
> >
> > ## Question 5: Explanation of Energy Increase and the Meaning of "fr"
> > We sincerely thank the reviewer for raising this important question regarding the apparent discrepancy between the small increase in $FLOPs$ and the significant increase in theoretical energy consumption shown in Table 4. We acknowledge that the formula used to calculate theoretical energy consumption was not explicitly mentioned in the main text. The energy calculation is based on the following structure, where not only $FLOPs$ but also increases in SNN time steps and changes in firing rate are directly reflected:
> >
> > $$SOPs=fr×T_s×FLOPs$$
> > $$E_{ANN}=E_{MAC}×FLOPs$$
> > $$E_{SNN}=E_{AC}×SOPs$$
> >
> > In response to this valuable observation, we have added a paragraph in the Appendix A.1 clearly explaining the above formula, the meaning of each term, and the cause of the energy increase. Furthermore, we confirm that the notation '$fr$' in Table 4 refers to the firing rate, and we have added a clear definition for this term. We appreciate the reviewer's careful attention to this detail.

---

### Official Review · Reviewer_3oyS · 2025-10-30

**Soundness:** 2
**Presentation:** 2
**Contribution:** 2
**Rating:** 2
**Confidence:** 3

**Summary:**

SummaryThis paper identifies a gap in SNN models for multivariate time-series forecasting: they typically focus on temporal dynamics and ignore inter-channel similarity. To address this, the authors propose a CSE module. This module first uses an attention-based clustering mechanism (adopted from ANN literature) to generate "soft" cluster membership probabilities ($M^{soft}$) for each channel. These soft probabilities are then converted into binary, spike-compatible "hard" cluster assignments ($M^{hard}$) using Bernoulli sampling and a Straight-Through Estimator (STE). Finally, this binary cluster information (a $K$-dimensional vector, where $K$ is the number of clusters) is concatenated along the SNN's time axis ($T_s$). The authors evaluate this method on six benchmark datasets using three SNN backbones (Spike-RNN, Spikformer, Spike-TCN) and report consistent performance improvements over the baseline SNNs.

**Strengths:**

- The paper correctly identifies an interesting and underexplored research gap: most SNN forecasting models process channels independently, ignoring inter-channel relationships that are known to be important in ANNs.


- The proposed CSE module is designed to be modular and is demonstrated to work with multiple SNN backbones (recurrent, transformer, and convolutional).


- The paper's presentation is a strength. The methodology is explained clearly, and Figure 1 provides a strong visual aid to understand the data flow.

**Weaknesses:**

-  The main appeal of SNNs is low energy consumption. This method, by extending the SNN time axis, increases computational load and leads to a 2.7x to 3.3x increase in theoretical energy consumption (Table 4). The authors' claim in the conclusion that the "accuracy-efficiency trade-off remains favorable" is unsubstantiated and appears incorrect. A ~300% energy cost for the reported accuracy gains is a terrible trade-off.
- The paper claims "consistent improvements," but the main results table (Table 1) and text (Section 5.2) show this is not the case. The method provides only "marginal" gains on ETTh1, "near-identical results" on METR-LA, and "slight degradations" on Electricity. This makes the method unreliable.
- The source of the energy increase is the integration method itself: concatenating $K$ static cluster vectors onto the $T_s$ time axis. This artificially inflates the sequence length that the SNN must process at every time step, which is an inefficient and naive way to inject static information. More efficient methods (e.g., using cluster IDs to modulate neuronal parameters, using a separate, smaller network to process cluster info) were apparently not explored.

**Questions:**

Please refer to my weakness part.

---

> ### Author Response · Authors · 2025-12-04
>
> ## Weakness 1: Energy Trade-off Validity and Justification
> We sincerely appreciate the reviewer's careful attention to the energy trade-off analysis presented in our work. We acknowledge that the proposed method incurs a theoretical energy increase of approximately 2.7–3.3 times compared to the SNN baseline, and we recognize that the term "favorable trade-off" used in the original conclusion did not adequately reflect this substantial increase.
>
> The primary goal of this study was not to achieve additional energy savings, but to enhance prediction accuracy by integrating channel similarity information into spike-form while maintaining the existing SNN structure. The time-axis extension design inevitably leads to increased energy consumption, as clearly shown in Table 4. However, when comparing ANN and SNN (baseline, ours) using the same theoretical energy model, the SNN family, including the proposed method, still exhibits lower absolute energy consumption than ANN.
>
> In response to the reviewer's valuable feedback, we have restructured Table 4 in the revision to enable clearer comparison between ANN and SNN, including metrics like firing rate $fr$, floating point operations $FLOPs$, SNN Timesteps $T_s$, $E_{MAC}$, and $E_{AC}$. Nevertheless, we agree that a roughly 3-fold increase compared to the baseline is not acceptable in all application environments. Therefore, the revised conclusion explicitly defines our method as "an option that can be chosen to improve accuracy when additional energy budget is available."
>
> Finally, we identified and corrected a scaling error in the energy (mJ) notation in the original Table 4. While the values themselves remain unchanged, we sincerely apologize for any confusion the incorrect notation may have caused.
>
> ## Weakness 2: Clarifying the Consistency of Performance Improvements Across Datasets
> We deeply appreciate the reviewer's concern regarding the use of "consistent improvements" in the main text. We acknowledge that this phrase may be an exaggerated expression that does not sufficiently reflect the marginal or near-identical performance observed in some datasets (e.g., ETTh1, METR-LA). In response to this insightful observation, we have revised the text to use a more accurate description (e.g., "improvements across most spike-based architectures and datasets" and "varying degrees of effectiveness").
>
> ## Weakness 3: On the Efficiency of the Proposed Correlation Integration Strategy
> We greatly appreciate the reviewer's thoughtful observation regarding the efficiency of the proposed correlation integration strategy. We acknowledge that directly concatenating channel cluster information along the time axis increases the time axis length, leading to higher theoretical energy consumption in the SNN. This study did not explore alternative integration methods such as threshold modulation, neuron-wise modulation, or lightweight auxiliary networks. In this regard, we recognize that the criticism regarding the structural simplicity of our approach is valid.
>
> However, we would like to clarify the rationale behind our design choice. The reason for choosing time-axis concatenation was to preserve the spike-form and avoid altering the existing SNN computation flow. Previous spike-form positional encoding studies [1,2] also employ time/feature axis concatenation for the same reason. Furthermore, in the comparative experiments of Section 5.3, our method achieved lower error (RRSE 0.660) than simple Time Extension or channel-axis concatenation, demonstrating the most stable performance among the compared approaches.
>
> Nevertheless, we fully agree with the reviewer that modulation-based methods or auxiliary network approaches hold significant potential for more efficiently reflecting channel structure and are considered meaningful as scalable future directions for this research. In response to this valuable feedback, the revision now clearly describes both the rationale for choosing time-axis concatenation and its limitations, including discussion of potential alternative approaches for future work in the Discussion section.
>
> [1] C. Lv, D. Han, Y.Wang, X. Zheng, X. Huang, and D. Li. Advancing spiking neural networks for sequential modeling with central pattern generators. In Advances in Neural Information Processing Systems (NeurIPS), volume 37, pp. 26915–26940, 2024a.
>
> [2] C. Lv, Y. Wang, D. Han, Y. Shen, X. Zheng, X. Huang, and D. Li. Toward relative positional encoding in spiking transformers. arXiv preprint arXiv:2501.16745, 2025.

---

### Official Review · Reviewer_sNRp · 2025-10-31

**Soundness:** 2
**Presentation:** 2
**Contribution:** 3
**Rating:** 2
**Confidence:** 2

**Summary:**

This paper introduces a spike encoding method that incorporates channel similarity awareness for multivariate time-series forecasting using SNNs. It addresses the limitation of existing SNN approaches, which primarily focus on temporal dynamics while neglecting inter-channel similarities. The proposed Channel-Similarity Encoding module employs attention-based clustering to compute soft channel memberships, converts them to binary spike forms via a Straight-Through Estimator, and integrates them into temporal spike encoding by concatenation along the SNN time axis. Testing on multi-domain datasets shows a decrease in RRSE, indicating the method's effectiveness.

**Strengths:**

1) The study adapts channel clustering concepts from ANNs to SNNs by proposing a framework that converts inter-channel similarity into a spike-compatible representation, utilizing the STE to binarize clustering results and align them with the SNN's sparse computation paradigm.

2) The method integrates with various SNN backbone architectures and encoding techniques, showing consistent performance improvements across multiple datasets from different domains, as supported by experimental results.

**Weaknesses:**

1) The results in Table 3 indicate that the optimal performance is achieved with n_cluster set to 3, yet the analysis lacks dataset-specific insights, such as visualizations of clustering outcomes, to substantiate the meaningfulness of clustering across different channels.

2) A notable concern is that when k = 1 (i.e., without clustering), the results still show a significant performance improvement compared to the baseline without clustering. This raises questions about the necessity and specific contribution of the clustering component.

**Questions:**

1) In the clustering process, does setting K=1 on the input equate to a no-clustering scenario, or does it effectively correspond to concatenating identical clustering information across all time steps (Ts) following the original sequence?

2) If so, why do the ablation results for K=1 deviate from the baseline, and could the authors provide a detailed analysis to elucidate the underlying factors contributing to this discrepancy?

---

> ### Author Response · Authors · 2025-12-04
>
> ## Weakness 1: Lack of dataset-specific justification for the effectiveness of clustering.
> We sincerely appreciate the reviewer's concern regarding the need for dataset-specific justification of clustering effectiveness. We acknowledge that the performance results in Table 3 alone did not sufficiently explain how meaningfully the clustering reflects the actual channel structure. In response to this valuable feedback, the revision includes a visualization of the channel similarity-based clustering results for the representative setting $K=3$ in the Appendix A.3. This visualization serves as evidence demonstrating that the clustering reflects actual relationships between channels to a certain degree, rather than being a simple parameter adjustment for this dataset. Although visualizations for all $K$ settings were not included due to space constraints, we believe that the provided example helps readers understand how the clustering results correspond to the channel structure of the data.
>
> ## Question 1: Clarification is needed on whether K=1 represents true no-clustering or simply repeats identical cluster information across all time steps.
> We greatly appreciate the reviewer's question regarding the nature of the $K=1$ setting. Indeed, $K=1$ is the "no-clustering" setting, as it does not perform clustering based on similarity between channels. However, we would like to clarify that $K=1$ is not identical to the baseline. While the baseline uses only the spike sequence generated by the spike encoder as input, $K=1$ adds a binary indicator in spike-form that all channels belong to a single cluster. This indicator is concatenated onto the time axis in the same format after the spike encoder output, effectively creating an additional representation space not present in the baseline. Therefore, $K=1$ should be understood as a setting that does not include clustering information but incorporates the spike-form indicator, resulting in time-axis augmentation.

---

> > ### Author Response · Authors · 2025-12-04
> >
> > ## Weakness 2 and Question 2: Explanation required for why the K=1 setting yields results that diverge from the baseline, including an analysis of the underlying factors.
> > We deeply appreciate the reviewer's insightful observation regarding the performance divergence between $K=1$ and the baseline. The performance gap arises from a structural difference in how the inputs are encoded. As the reviewer correctly notes, $K=1$ constitutes a no-clustering setting because all channels belong to a single group. However, $K=1$ is not equivalent to the baseline, which uses only the spike encoder's temporal outputs. In the $K=1$ case, a binary indicator denoting membership in the single cluster is appended in spike form, providing an additional temporal input stream that is absent in the baseline.
> >
> > This structural difference has an important consequence: while simple time-axis extension of the spike encoder output often results in certain channels never firing throughout the entire window (meaning their information is not propagated to subsequent layers), our $K=1$ integration ensures that every channel produces at least one spike. This enables all channels to contribute information even when the original temporal encoder would have produced no activity. This mechanism explains why $K=1$ achieves modest improvements over the baseline despite not incorporating channel similarity information.
> >
> > ## Question 3: Concerns about over-partitioning and empty clusters when K≥4
> > We sincerely appreciate the reviewer's attention to the over-partitioning phenomenon observed when $K \geq 4$. The reviewer raises an important question about whether this issue reflects an intrinsic property of the data or a methodological limitation. We acknowledge that our original discussion did not sufficiently address this distinction.
> >
> > In our analysis of the Electricity dataset, we observed that when $n_{cluster} \geq 4$, empty clusters (clusters to which no channels are assigned) emerge, and similar tendencies appear in other datasets when $n_{cluster}$ exceeds 3 or 4. Rather than suggesting an intrinsic "optimal number of groups" inherent to the data, this finding indicates that when the number of clusters exceeds a certain threshold, the stability of the spike-form integration process degrades, leading to reduced forecasting performance. This over-partitioning destabilizes channel clustering and dilutes the structural information conveyed through spike-form indicators.
> >
> > In response to this valuable observation, we have revised the main text to clearly distinguish between (i) the data-driven nature of channel clustering patterns and (ii) the methodological limitations that arise when cluster numbers become excessive, and we have added a dedicated discussion to explicitly address these points. The revision now explicitly states that the performance degradation at $K=4$ stems from over-partitioning in the spike-form integration process rather than from discovering a "natural" cluster structure in the data. We believe this clarification better reflects the experimental findings and prevents potential misinterpretation of the results.

---

### Official Review · Reviewer_ioK9 · 2025-11-02

**Soundness:** 3
**Presentation:** 2
**Contribution:** 2
**Rating:** 4
**Confidence:** 3

**Summary:**

The paper introduces a method to improve multivariate time-series forecasting in Spiking Neural Networks (SNNs) by incorporating inter-channel similarity. The core idea is that existing SNNs focus mostly on temporal dynamics and process channels independently, missing out on valuable cross-channel information. The proposed Channel-Similarity Encoding (CSE) module first uses an attention mechanism to cluster channels based on their similarity, producing soft cluster assignments. These soft assignments are then converted into binary, spike-compatible representations using a Straight-Through Estimator (STE). Finally, this channel information is concatenated with the standard temporal spike encoding and fed into various SNN backbones. Experiments across six datasets show that this method consistently reduces forecasting error compared to baseline SNNs, albeit with an increase in computational and energy costs.

**Strengths:**

1. The paper addresses a clear and sensible gap in the literature. The motivation, that SNNs for time-series have largely ignored inter-channel relationships, a proven benefit in the ANN world—is well-argued and provides a strong foundation for the work.
2. The proposed method is technically sound and modular. Using an attention-based clustering mechanism is a reasonable approach, and the use of the Straight-Through Estimator (STE) to integrate this into a spike-based framework is appropriate. The design allows it to be plugged into different SNN backbones (RNN, Transformer, TCN), which is a nice feature.
3. The experimental validation is reasonably comprehensive. The use of six standard benchmarks and three different types of SNN architectures demonstrates the general applicability of the proposed CSE module. The inclusion of an analysis on theoretical energy consumption is also commendable, as it provides a more complete picture of the trade-offs involved.

**Weaknesses:**

1. The trade-off between accuracy and efficiency needs more discussion. The results show a relative RRSE reduction of around 5.2% on average (Line 299), but Table 4 indicates a significant energy consumption increase of 2.7x to 3.3x. This is a substantial cost for a modest accuracy gain and seems to undermine the primary motivation for using SNNs, which is their efficiency. The paper presents these numbers but doesn't really delve into the implications of this trade-off.
2. While the method improves upon SNN baselines, the performance still lags considerably behind standard ANN models. For instance, in Table 1, the proposed method with Spike-TCN on the Electricity dataset (horizon 96) achieves an RRSE of 0.350, whereas the iTransformer baseline achieves 0.226. This is a significant performance gap. A more direct acknowledgment and discussion of this remaining gap would help contextualize the contribution.
3. The justification for the specific design choice of concatenating cluster information along the time axis could be stronger (Section 4.4). The paper mentions it is done to "preserve binary spike values" (Line 258), but it's not immediately obvious why this is superior to other potential fusion mechanisms. An ablation study or a more detailed explanation of alternatives considered would strengthen this part of the methodology.

**Questions:**

1. Regarding the accuracy-energy trade-off highlighted in Table 4: could the authors elaborate on the practical applications or scenarios where a ~3x increase in energy cost is justified for the performance gains observed? Given that SNNs are often targeted for resource-constrained environments, this seems like a critical point to address.
2. In Section 4.4, the binary cluster memberships are replicated and concatenated along the SNN time axis. Have other methods for integrating this information been considered? For example, could the cluster embeddings be used to modulate the firing thresholds of neurons or act as a form of attention mechanism over the temporal spikes? A little more insight into the design process here would be helpful.
3. The clustering process described in Section 4.2 seems to generate a single set of cluster assignments for the entire input time-series window. This implies an assumption that inter-channel relationships are static within that window. Was any thought given to capturing more dynamic relationships, where channels might cluster differently at different points in time?
4. The parameter analysis in Section 5.4 and Table 3 shows that performance is quite sensitive to the choice of n_cluster. Specifically, performance degrades when moving from 3 to 4 clusters. Is there an intuition for why this might be the case? Does this suggest that the datasets used have a small number of inherent, dominant channel groups?

---

> ### Author Response · Authors · 2025-12-04
>
> ## Weakness 1: Insufficient discussion of the accuracy–energy trade-off.
> We sincerely appreciate the reviewer's concern regarding the accuracy–energy trade-off analysis. We acknowledge that the proposed method achieves approximately a 5.2% RRSE improvement while incurring a theoretical energy increase of 2.7–3.3 times compared to the SNN baseline. The objective of this study was not to further enhance the energy efficiency of SNNs, but rather to improve accuracy by integrating channel similarity information in spike form while maintaining the existing SNN architecture. We recognize that this point was not sufficiently emphasized in the main text.
>
> Nevertheless, as shown in the revised Table 4, even with increased energy consumption compared to the baseline, the proposed method still maintains significantly lower energy consumption than ANN models (RNN, TCN). Thus, while it may not be suitable for extremely energy-constrained environments, it can be a practical choice for SNN-based applications where additional energy budget is permitted. In response to the reviewer's valuable feedback, the revision clearly describes the significance of this accuracy–energy trade-off and its applicable scenarios in the Discussion section.
>
> ## Weakness 2: Large performance gap vs. ANN models remains.
> We greatly appreciate the reviewer's observation regarding the performance gap between our method and ANN models. We acknowledge that while our proposed method improves performance over SNN baselines, the performance gap with ANN models such as iTransformer remains significant in certain settings. This stems from the structural differences between dense computation-based ANNs and event-driven SNNs, which is a limitation commonly observed in existing SNN-based prediction models.
>
> The goal of this research is not to replace ANN performance, but to enhance the performance within existing SNNs while preserving the sparse binary computation structure of SNNs. We recognize that our research focus was not sufficiently stated in the paper. In response to this feedback, we have more clearly described the reasons for the performance gap compared to ANNs and the scope of this research's contributions.
>
> ## Weakness 3: Justification for time-axis concatenation is unclear.
> We deeply appreciate the reviewer's concern regarding the rationale for selecting time-axis concatenation. We acknowledge that the justification for this design choice and its comparison with alternative fusion methods were not sufficiently explained in the original manuscript. While this study did not explore threshold modulation or attention-based methods, it experimentally compared two structural alternatives (SNN time-axis extension and channel-axis extension).
>
> These approaches exhibited repeated instances where the spike encoder output for a specific channel throughout the entire window, resulting in ineffective transmission of channel similarity information. In contrast, the proposed time-axis concatenation directly integrates cluster membership as a spike-form binary indicator into the temporal structure. This enables channel structural information to be conveyed independently of the spike encoding generated by the temporal encoder. We believe this structural stability led to the performance advantage demonstrated in Section 5.3. In the revision, we have added clearer explanations of this rationale and the experimental comparisons.

---

> > ### Author Response · Authors · 2025-12-04
> >
> > ## Question 1: When is a ~3× energy increase acceptable?
> > We sincerely appreciate the reviewer's important question regarding the acceptability of the energy increase. We fully acknowledge that the proposed method results in approximately three times the theoretical energy increase compared to SNN benchmarks, which represents a clear trade-off and cannot be justified in all deployment scenarios. The objective of this study was not to achieve extreme energy savings, but rather to improve accuracy by integrating channel structure information while maintaining the existing SNN architecture. We recognize that this point was not sufficiently emphasized in the paper.
> >
> > Nevertheless, as shown in Table 4, the proposed method still exhibits significantly lower absolute energy consumption compared to ANNs (such as RNNs and TCNs). Therefore, it can be a practical choice in SNN-based scenarios where a certain level of additional energy consumption is permissible. This trade-off may be reasonable, particularly in applications where higher accuracy than that of the baseline is crucial, such as safety-critical predictions or high-confidence anomaly forecasting. In response to this valuable question, the revision now explicitly discusses the applicable scenarios and limitations of this trade-off.
> >
> > ## Question 2: Were other fusion strategies considered?
> > We greatly appreciate the reviewer's suggestion regarding alternative fusion strategies. The threshold modulation or temporal attention-based integration methods proposed by the reviewer are indeed viable alternatives. We acknowledge that this study did not directly explore these approaches; instead, it compared two structural strategies (SNN time-step expansion and channel-axis concatenation) to evaluate differences in channel similarity integration methods.
> >
> > The reason for choosing time-axis concatenation is that it can reliably transmit channel information while preserving spike-form, and this design is also inspired by prior studies [1,2] that concatenate auxiliary spike-form positional encodings into existing temporal spike encodings. Indeed, Table 2 shows that this method achieved the lowest RRSE compared to the two baselines. Nevertheless, we fully agree with the reviewer that exploring modulation-based or attention-based fusion strategies represents a meaningful direction for future work. The revision now includes discussion of these alternative approaches as potential future directions.
> >
> > [1] Lv, C., Han, D., Wang, Y., Zheng, X., Huang, X., & Li, D. (2024). Advancing spiking neural networks for sequential modeling with central pattern generators. Advances in Neural Information Processing Systems, 37, 26915-26940.
> >
> > [2] Lv, C., Wang, Y., Han, D., Shen, Y., Zheng, X., Huang, X., & Li, D. (2025). Toward Relative Positional Encoding in Spiking Transformers. arXiv preprint arXiv:2501.16745.
> >
> > ## Question 3: Static window-level clustering may be limiting.
> > We sincerely appreciate the reviewer's insightful observation regarding the static nature of our clustering approach. We acknowledge that the clustering in this study generates a single static cluster assignment for the entire input window. This follows the window-level channel modeling approach commonly used in recent multivariate forecasting studies such as DUET [1] and CCM [2].
> >
> > However, we fully agree with the reviewer that channel relationships may change over time, and addressing dynamic clustering or time-varying dependencies within the SNN structure is represents a valuable direction for future work. In response to this valuable feedback, the revision now explicitly discusses this limitation and potential future directions toward dynamic clustering approaches.
> >
> > [1] Qiu, X., Wu, X., Lin, Y., Guo, C., Hu, J., & Yang, B. (2025, July). Duet: Dual clustering enhanced multivariate time series forecasting. In Proceedings of the 31st ACM SIGKDD Conference on Knowledge Discovery and Data Mining V. 1 (pp. 1185-1196).
> >
> > [2] Chen, J., Lenssen, J. E., Feng, A., Hu, W., Fey, M., Tassiulas, L., ... & Ying, R. (2024). From similarity to superiority: Channel clustering for time series forecasting. Advances in Neural Information Processing Systems, 37, 130635-130663.

---

> > > ### Author Response · Authors · 2025-12-04
> > >
> > > ## Question 4: Why does performance drop when n_cluster increases?
> > > We deeply appreciate the reviewer's question regarding the performance drop when n_cluster increases. In the Electricity dataset analyzed in this study, we observed that empty clusters (clusters with no assigned channels) occurred when $n_{cluster} \geq 4$, and a similar trend was observed in other datasets when $n_{cluster}$ was 3 or higher, or 4 or higher. This over-partitioning destabilizes channel clustering during the propagation of spike-form indicators. This pattern is also visually evident in Figure 4.
> > >
> > > Rather than suggesting an intrinsic "optimal number of groups" inherent to the data, we believe this performance degradation should be understood as a tendency for expression stability to decrease during spike-form integration when the number of clusters exceeds a certain threshold. In response to this valuable question, the revision now clearly distinguishes between data-driven clustering behavior and methodological limitations, explicitly stating that the degradation stems from over-partitioning in the spike-form integration process.

---

### Meta-Review · Area_Chair_JhQW · 2026-01-07

**Summary:**

All reviewers agree the paper identifies a clear gap—SNNs for multivariate forecasting ignore inter-channel similarity—and propose a modular plug-in (CSE) that gives 3–7 % RRSE drops on six datasets. The chief objection is the cost: 2.7-3.3× theoretical energy increase for these modest gains, undermining SNN’s main selling point. Two reviewers also note remaining large gaps vs. ANNs and question the static, concatenate-along-time integration strategy. Rebuttal clarifies goals (accuracy, not efficiency), adds ablations/visuals, but does not remove the energy penalty.

**Reviewer Concerns:**

Reviewer Concerns Addressed
• Clarification of trade-off scope and absolute energy still below ANNs.
• K=1 ablation explanation and empty-cluster issue.
• Static clustering limitation acknowledged as future work.
• Writing/citation/formatting errors.

Still Outstanding
• No solution for the 3× energy increase; method remains inefficient for edge deployment.
• Performance still far below ANNs (≈ 35 % higher RRSE on Electricity).
• No dynamic or more efficient fusion schemes tested; concatenation along time axis is still naïve.

**Reviewer Scores:**

ioK9: 4 → 5 (borderline)
sNRp: 2 → 3 (weak reject)
3oyS: 2 → 3 (weak reject)
YQYf: 2 → 3 (weak reject)

---

### Decision · Program_Chairs · 2026-01-26

Reject